# Minimally Monophyletic Genera Present within Meso- and Macrogenera

Richard H. Zander [ID]

Missouri Botanical Garden, 4344 Shaw Blvd., St. Louis, MO 63110, USA; rzander@mobot.org

**Abstract:** Past efforts to identify and characterize minimally monophyletic groups (microgenera) by deconstructing larger bryophyte genera successfully determined 10 microgenera comprising the moss family Streptotrichaceae. Thirty other microgenera have also been found in the moss family Pottiaceae. A microgenus consists of one ancestral species and, optimally, four immediate descendant species, each of which shares exactly the same ancestral traits. To determine if microgenera were common, evidence of these in larger genera was garnered from published estimates of species per genus in other groups and from molecular cladograms in the moss family Pottiaceae. Both classical mesogenera and cladistically enlarged macrogenera exhibited an internal granularity of one to five species, either as multiple species below the inflection point in the hollow curve of logarithmic graphs of species per genus or as small groups of molecular cladogram branches. Microgenera are basic units of evolution. The constancy of size and monothecy of traits in microgenera give them properties that larger taxonomic groups lack. Sequences of microgenera monophyletic are easily concatenated, adaptational changes may be directly determined, self-similarity across scale allows extended scientific inferences, and traits can be associated with survival across millions of years of environmental perturbation.

**Keywords:** ancestron; evolution; macrogenus; mesogenus; microgenus; novon; self-similarity; structured monophyly

## 1. Introduction

A series of papers, particularly two recent works [1,2], introduced the concept of the minimally monophyletic genus (MMG), consisting of one morphological ancestral species and usually one to four descendant species where four immediate descendants are optimal. The MMG was dissilient, exploding like extant evidence of extant punctuated equilibrium, monothetic, with a single diagnosis for each species, and may be concatenated into a fully monophyletic lineage of several successive genera in depth of geologic time [2]. This models an evolutionary tree of structured monophyly, a caulogram. From NK-analysis (N is nodes, K is inputs) in the context of random Boolean network modeling, four descendant species were found to be optimum in competition and mutualism [3,4], and such was demonstrated in studies reducing classical large genera to their MMG units [5]. The MMG was demonstrated to be strongly supported by Bayesian statistics using second-order Markov chains and conjugate priors [1,5]. The lineage of several MMGs is expected to retain ancient traits valuable in surviving recurring environmental perturbations over geologic time [2].

The optimality of four traits (in addition to four descendants) implies self-similarity across scales and that evolution is fractal at dimension 1.16 [6]. Dynamic evolutionary relationships of morphological traits in MMGs, here termed *microgenera*, are as follows: The *novon* is the set of new traits obtained from ancient traits of the ancestral species, usually four in number and different between the descendants. The *ancestron* is the set of ancestral traits, of which the *immediate ancestron* consists of the new traits obtained from its own ancestor and passed on without change to each descendant species. The *reserve ancestron*

includes the more ancient traits commonly of little survival value at present but modified through state changes to make up the novons of descendants. It is hypothesized that the immediate ancestral protects the descendant species sympatrically, and the novon is active in exploring new habitats. Such stability of the process can be supported by natural selection [7] or the "species selection" of Stanley [8]. It is possible that the immediate ancestron acts in stabilizing selection, while the radical novon versus the conservative reserve ancestron provides balancing selection.

Major concepts of the taxonomic rank of genus here delineated are the microgenus, mesogenus, and macogenus. Their origins and uses in modeling and understanding processes in nature are evaluated. The *mesogenus* is the standard concept and consists of variously sized groups of similar species that are related by the apprehension of overall similarity plus traits considered conservative and by explanations of trait similarity in classical evolutionary theory. It may be monophyletic or paraphyletic, and larger genera are generally polythetic. Evolutionary relationships are modeled as branching sets of large and somewhat heterogeneous groups, as with the Besseyan cactus [9,10].

The *macrogenus* consists of multiple mesogenera gathered together under the umbrella of strict phylogenetic monophyly. It is made monophyletic by combining many mesogenera and is necessarily polythetic. It is modeled by a cladogram with taxa terminal on the branches. The macrogenus may be based on morphological data, with relationships through estimated shared morphological ancestors, or on molecular data, in which taxonomic units are grouped through shared molecular ancestors.

The *microgenus* consists of one ancestral species and a few descendant species strongly related by a set of traits each shares with the ancestor and with each other. It is the minimally monophyletic evolutionary unit (MMG) and is also monothetic such that all species have one diagnosis, usually of two to four traits. It is modeled as the basic building block in a caulogram of structured monophyly, showing a concatenated and branching series of microgenera. It follows complexity-based rules [11] in taxonomy associated with an evolutionary rule of four [1,6,12], where four is the optimum number of descendant species and of novon traits. This rule is not evident in other concepts of genera but may be indicated in the preponderance of genera with one to five species and in similar granularity in molecular cladogram, as investigated here. Given that cladistic trees present a cluster analysis of synapomorphies, limited to common descent without identifying extant ancestral species, the monophyly modeled in a cladogram is unitary and internally unstructured.

Since Aristotle [13] introduced species and genera as hierarchical units for grouping organisms (and much else in logic and metaphysics), the practice of grouping species into large sets has been problematic. There has been no clear-cut standard method of doing so, and different taxonomists commonly revise generic limits to reflect apparent group similarities and gaps. Given no agreed-upon definition of the word genus, variation in concepts goes beyond simple reflection of new knowledge. Various ideas have been advanced beyond the direct apprehension of natural supraspecific groups by knowledgeable taxonomists, including numerical taxonomy grouping by shared similarity [14] and cladistics analysis by shared synapomorphies [15]. The standard taxonomic practice may produce supraspecific groups that are not monophyletic (may have embedded lesser monophyletic groups) and are polythetic (inclusive of multiple ancestral species such that species do not all share one universal diagnosis). Cluster analysis assumes shared similarity must reflect much monophyly and much monothecy [16,17] but without providing details of descent. Cladistics ensures monophyly by grouping all species with shared ancestry and not naming embedded monophyletic groups as different genera, but monothecy is not expected; that is, the genus may not have a single diagnosis. A new taxonomy-based technique is now available for recognizing microgenera as sets of species that are minimally monophyletic, that is, having one ancestral species, and it turns out these are also monothetic, having one diagnosis.

Macroevolutionary analysis, as presented here and in past papers (Zander, see above), uses taxonomic traits as evolutionary information. These are expressed traits that may also

include other easily observed features unique to the species, such as habitat, traits that are clearly not part of or are a constant effect of another trait. One might argue that there are many traits in addition to traits of taxonomic value or obvious unique ecological association, and such should also be used in evolutionary analysis. Taxonomically valuable traits are, however, the focus of this method of evolutionary analysis. There are doubtless characters at some level of organization that might be loaded with much the same information as a trait at another level through pleiotropy or a direct causative connection. A leaf may be green, it may have chlorophyll in it, a molecule may be one with iron in it, and a gene or set of genes may code for all of these as one evolutionarily informative character state. The burgeoning number of traits causing the same process or effect would violate the empirically derived evolutionary rule of four taxonomic traits as optimum for a species. Such portmanteau traits are indeed informative, but only as they are distinct from other traits at the same level of organismal organization.

## 2. Materials and Methods

The classical basic unit of taxonomy and classification is the species, which is the main focus of evolutionary theory. A recent study has suggested, however, that genera may be defined in some cases rather rigorously as minimally monophyletic units and, as such, have unique and complex properties that allow a theoretical window on processes of evolution that occur at the lineage and ecosystem levels. The objective of the present study is to investigate the internal structure of micro-, meso-, and macrogenera as possibly overly complex units in evolutionary analysis. In particular, an important question is if any microgenus structure is evident in the classifications and evolutionary models associated with meso- and macrogenera. The method chosen here is to find evidence of microgeneric structure in evolutionary studies of classical taxonomy and of granularity of appropriate size in the cladograms of molecular phylogenetics.

Examples in the literature of the evaluation of species per genus were compiled, particularly those presenting logarithmic graphs. Two molecular studies discussed below were examined for granularity at the level of microgenera. Previously cited examples of molecular paraphyly are reinterpreted as evidence of molecular strains of surviving ancestors of microgenera. Illustrations were made to clarify evidence of previously hidden information on the internal structure of meso- and macrogenra.

## 3. Results

### 3.1. Evolutionary Models of the Mesogenera of Classical Systematics

The genus name is a central feature of mesogenera, being popularized by Linnaeus with his binomial naming construct and formalized in a series of botanical codes of nomenclature. It is the principal taxonomic rank above species, beyond which there is no standard definition. The Code [18] simply asserts (Art. 3.1, Note 1) that species must be assigned to genera.

Most post-Linnean taxonomic works group species into genera and higher ranks, a hierarchical sorting that allows great convenience [19] in reporting taxonomic, ecological, and floristic research. At least in modern times, the composition of genera avoids, when possible, artificial groupings in favor of inferred evolutionary relationships [8,20]. That is, by a combination of ingroup similarity, apparently due to shared ancestry, and gaps between groups due to genetic isolation and adaptive divergence. Classical taxonomic usage is summarized by Clayton [21]. Evolutionary diagrams are generalized, exemplified by the well-known "Besseyan cactus" [22]. This branching diagram has been condemned by phylogeneticists [23–25] because the taxa are polarized as primitive and advanced, and the method is not repeatable—more generally because taxa were derived from other taxa rather than related solely by shared ancestry terminal to a cladogram. Additionally, the genus itself may not be clearly monophyletic as shown in a cladogram, it is too difficult to determine advanced and derived traits, no biological theory is involved, delimitation is notional and circular, and general purpose classifications are basically ambiguous [26,27].

Large genera are known to have considerable granularity in their makeup, usually expressed as subgenera or taxonomic sections. Graphing the number of species in a large set of genera can provide some information on possible internal structuring along the lines of minimally monophyletic genera. Stevens [28] graphed the size distribution of genera of vascular plants in the work of Bentham and Hooker. This was a hollow curve with most genera having one to five species, and the sharpest flexion of the species per genus curve was at about five species per genus (Figure 1A). A hollow curve commonly follows a power law, where graphing on log–log axes yields a straight line [29], but that may not always be the case.

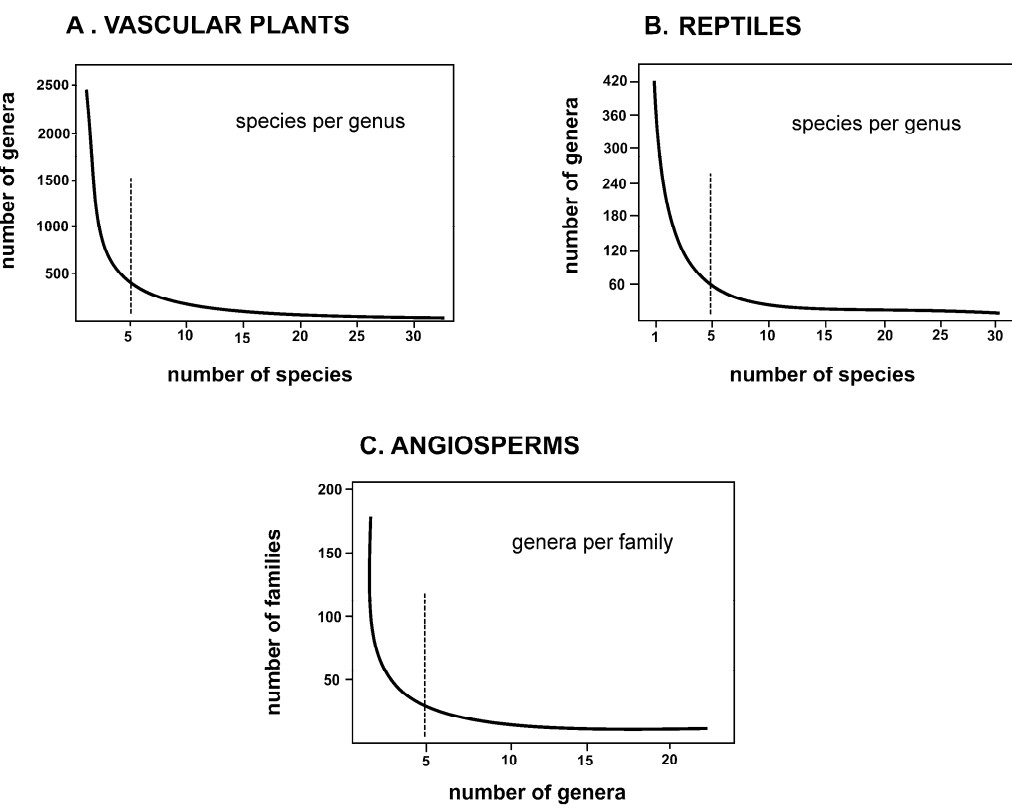

**Figure 1.** (**A**,**B**) Hollow curve graphs of numbers of species in genera. Flexion of curve at five species per genus implies multiple mesogenera with hidden structured microgenera: (**A**) Vascular plants. (**B**) Reptiles. (**C**) Angiosperms, genera per family, where identical flexion at five genera per family implies self-similarity across scales, including species per genus.

François [30] introduced a stochastic model of species generation. Using reptiles as an example, analysis of species per genus produced a hollow curve (Figure 1B) quite like that of Stevens'. According to François, Reptilia consists of almost 11,000 species in 1196 genera, averaging 9.1 species per genus. The hollow curve of frequencies of numbers of species per genus likewise bends most sharply at five species per genus. Thus, although the average number of species per genus is about nine for reptiles, most genera likewise have one to five species, and the fewer genera with greater numbers of species level off as asymptotes. Microgenera, in like manner, follows an evolutionary "rule of four", that is, of optimally four immediate descendant species for each ancestral species [2,6], where gradual extinction through geologic time trims the optimum, leaving most genera with a range of one to five species.

The hollow curve is well matched in the study of Clayton [31], who discussed the logarithmic distribution of genera in angiosperm families. The same flexion is demonstrated at five genera per family (Figure 1C). This augurs for the evolutionary rule of four (four descendants per ancestor) being self-similar across scales [6,32,33].

A world monograph of the large genus *Hypericum* by Norman Robson [34] found that the ca. 470 species were distributed in 36 sections, averaging about 13 species per section. An evolutionary diagram showed three nexuses, with six, seven, and eight main branches, respectively, plus many secondary branches. The largest genus of flowering plants is *Astragalus*, with ca. 3000 species. A recent revision of the group for North America north of Mexico, including Greenland [35], found 357 species grouped in 94 taxonomic sections, yielding 3.8 species per section on average.

A phylogenetic study of biodiversity [36] found evidence of well-structured and coherent biological entities above the rank of species but looked for evolutionarily significant units and did not evaluate species per genus.

It may be suggested that the limitation of minimally monophyletic genera to optimally one ancestor and four descendants is due to a bias. This bias may be internal to the taxonomist, who may group by the handful, that is, by easily recognized convenient sets [19], or external by some process in nature amenable to complexity theory, e.g., natural selection at the genus level [6,8]. Given multiple instances of a rule of four, detected by different researchers and at different scales, the latter case seems more likely.

### 3.2. Evaluation of Macrogenera

3.2.1. Morphologically Based Cladistics

Numerical or computational systematics [14] is also called phenetics because it originally operated with morphological characters. It promoted gaps and similarity for the working definition of a genus, stating that similarity alone ensures a degree of monophyly and that classification is automatic and repeatable. Cladistics, emphasizing grouping by shared synapomorphies as evidence of shared ancestry, found no way to distinguish the genus beyond that of a set of clades following the principle of strict monophyly. This is regularized in the PhyloCode [37], where the genus is a category considered independent of the cladistic purview. The cladogram, however, is a fixed dichotomously branched tree (multifurcate only when "poorly resolved") and was originally intended as a tabula rasa for cluster analysis, not as an actual model of evolution.

The low support values of nonparametric bootstrapping have been unfairly compared with the high statistical support for molecular studies using Bayesian or maximum likelihood techniques. The data set in morphological cladistics, however, is not appropriate for the statistical method of the bootstrap [38,39]. The data set, as commonly used, is actually a set of descriptions of taxa, and subsampling descriptions naturally yield results with large lacunae and low bootstrap values. The data should properly be the traits of the specimens, not the taxa, and bootstrapping should find that subsampling the specimens (with attached descriptions) gives high bootstrap values.

Van Valen [40] pointed out that extant ancestral taxa exist quite commonly. Yet cladistics places all taxa as a terminal on a cladogram, a dichotomously branching tree. Consider a set of 10 closely related species. To demonstrate shared ancestry, logically, there should be an inferred shared ancestor between every parsimoniously optimized split on the cladogram. But, there is no room for nine shared ancestors with intermediate traits, and common ancestry either takes on a metaphysical dimension or reverts to cluster analysis by synapomorphies. This is complicated by the fact that ancestral taxa are not distinguished from descendant taxa, and all are terminal on a cladogram. Clearly, a genus based on ancestor–descendant relationships is difficult to demonstrate in a cladogram.

The mesogenus has been of much use in evolutionary systematics, such as in works by Simpson [41]. He suggested (p. 32) that a measure of the rate of evolution would be to divide the number of genera in a sequence by the duration of that sequence's existence. In the lineage of horses, eight successive genera lasted about 60 million years, one genus apparently evolving into the next; thus, these genera lasted on average 7.5 million years. This contrasts sharply with the estimate of 22 million years for the most recent genus of Streptotrichaceae [2], with other genera up to four in depth of geologic time originating much more distantly in the past and still extant. In reference to confusing multimodal

graphs of morphological change over time, Simpson (p. 57) wondered if there might not be an underlying rule or law of evolution that contributes to an unimodal scaling of such change. The average number of species in the eight genera of horses, including the extant *Equus* is 4.6. The average number of species per genus among the mammals as a whole, a well-studied group, has been demonstrated to be 4.4 [42], perhaps reflecting an evolutionary rule of four.

A molecular study [43] influenced a morphological revision of *Oxystegus* and *Pseudosymblepharis* and allied genera [44] such that these and other classically recognized genera were lumped under the name *Chionoloma*. The morphological study was otherwise entirely standard and well accomplished, and the key to species mostly split the species along classical mesogeneric lines (some species with anomalous traits were, as standard practice, eliminated early in the key). There were 22 species recognized. Summaries of the salient distinguishing traits given in the discussions indicated that there was an average of 4.3 distinctive morphological traits per species, with a range of three to six traits and a mode of 5. If this number is a valid indication of the number of new traits distinguishing a species from its immediate ancestral species, then the evolutionary rule of four new descendants per ancestor may also be valid for traits and self-similar across scales.

### 3.2.2. Granularity in Molecular Cladograms

Granularity in molecular cladograms implies multiple minimally monophyletic groups subsumed in a large, phylogenetically holophyletic genus. Evolutionary analysis is most productive at the species level (population genetics, ecology of species, etc.) because mesogenera are less definable, commonly paraphyletic, and may experience multiple evolutionary processes associated with different internal groupings of species. Macrogenera are even more massive agglomerations of species, all given the same genus name through the implementation of the cladistic principle of holophyly, and unique results of evolutionary processes applicable to all species in the macrogenus must be few. Microgenera, if comprising the bulk of macrogenera, would be exact reconstructions of an evolutionary process involving the shared intermediate ancestron and have a clear footprint in geological time [2].

I have pointed out that microgenera are supported by high Bayesian posterior probabilities (BPP), with alternative branch arrangements supported by conjugate priors, meaning all probabilities add to 1.00. Markov chain Monte Carlo (MCMC) methods of estimating Bayesian posterior probabilities deal with complex molecular data with a shortcut involving multiple sampling and analysis of the data, which gives an approximation of the BPP. MCMC studies actually give a normalized solution such that alternative branch arrangements are, in fact, of zero statistical certainty when posteriors are given as 1.00, or statistical certainty. This is because posterior probabilities are calculated with the prior odds ratio for values with the same marginal probabilities, and only the prior and likelihood are then relevant [45].

MCMC reconstructions may then be considered very accurate in interpreting the data set as evolutionary orderings of molecular sequences. Dealing with the OTUs as a species, however, involves a problem. Molecular analysis cannot deal with the fact of extant (not fossil) punctuated equilibrium under the constraint of fractal evolution involving the rule of four (optimally four descendants, optimally four traits in the novon). Short branches in a phylogram do imply a burst of speciation, but this is not pursued here. The molecular cladogram does show the order of descent, but the results must be translated into a multichotomous caulogram with ancestral species at the nodes. In a given study, about half the species are ancestors or ancestors of ancestors, and only half are terminal taxa [1]. Thus, half the terminal taxa in a molecular cladogram also occupy the nodes. In a morphological cladogram of closely related species, it is clear that there is no room for unknown intermediate shared ancestors, but the large data space in a molecular data set obviates this problem, which, however, continues to confound translation to a stem-taxon caulogram. A phylogenetic work-around for this is to simply map the morphological

traits to the molecular cladogram, but this pushes the problem back to the fact that the morphological traits create very well-supported evolutionary trees by themselves, which are incongruent when presented as a morphological evolutionary tree, never used as a prior.

The order of descendants is not particularly important when punctuated equilibrium is invoked, where the more closely in time that descendant species are generated, the more likely they each are to find uninhabited substrates/niches [6]. The order of primary ancestors in a genus is important as it allows the comprehensive construction of a lineage model given the overall breadth and trajectory of the entire assemblage of species traveling through time. The lineage has a unique contribution to the survival of an ecosystem, and together, these space-time-ship Earths temper entropic balance, enhancing robust survival across epochs of repeated environmental perturbations [2].

Two molecular cladograms (Figure 2A,B) were examined for evidence of internal minimally monophyletic structure. These were both macrogeneric constructs combining several classically distinguished mesogenera into larger units that were molecularly monophyletic in the strict, holophyletic sense. The cladograms are reproduced here with terminals ending in samples of the same species connected with a vertical bar.

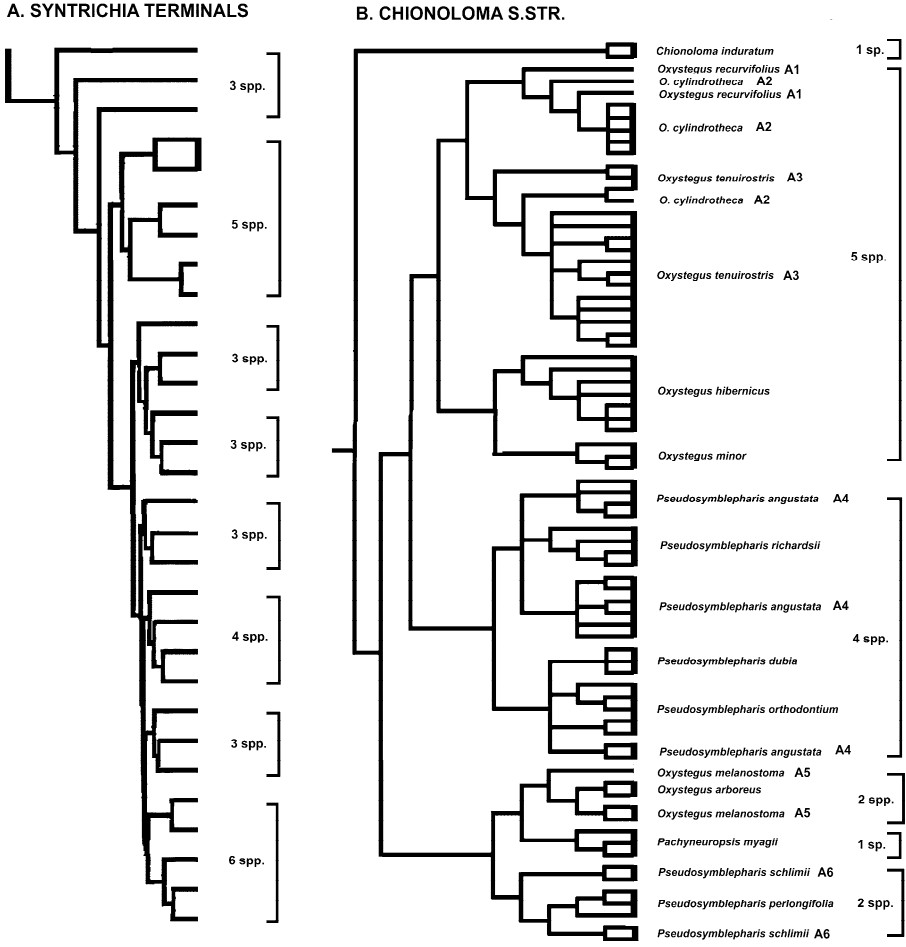

**Figure 2.** Molecular cladograms with granularity imply hidden morphological microgenera within macrogenera. (**A**) Terminal portion of the genus *Syntrichia* s.lat. cladogram. (**B**) *Chionoloma* s.lat. reconstructed as constituent classical mesogenera, which are small enough to probably prove to be microgenera, perhaps with secondary ancestry (descendants originating descendants).

*Syntrichia* Molecular Cladogram

The large moss genus *Syntrichia* Brid. was even more greatly enlarged in a molecular study [46] with the addition of *Calyptopogon* (Mitt.) Broth., *Sagenotortula* R. H. Zander,

*Streptopogon* Wils. ex Mitt., and *Willia* Müll. Hal. These small genera segregated well as coherent subclades but were embedded among *Syntrichia* species, thus open to synonymy at the genus level following the phylogenetic classification principle of strict monophyly. The cladogram also demonstrated many subclades of *Syntrichia* s.str. A terminal portion of the molecular cladogram is reproduced here (Figure 2A), demonstrating that potential microgenera in *Syntrichia* s.str. show themselves as eight small clades molecularly. The number of species in each molecular subclade includes three to six species, totaling 30 species, averaging 3.75 species per molecular subclade. If the three long branches at the base of the partial cladogram (Figure 2A) are actually unispeciate microgenera, then there are 10 subclades, and the number of species per microgenus is 3.00—there is no way of telling this short of conducting a morphological ancestor–descendant study. The actual identification of a microgenus requires the identification of a single ancestral species. The difficulty of finding the exact placement of an ancestral species on a molecular cladogram can be graphically demonstrated (Figure 3A–C).

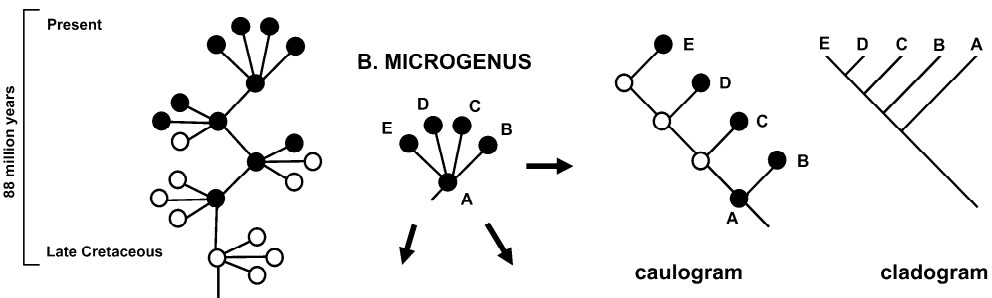

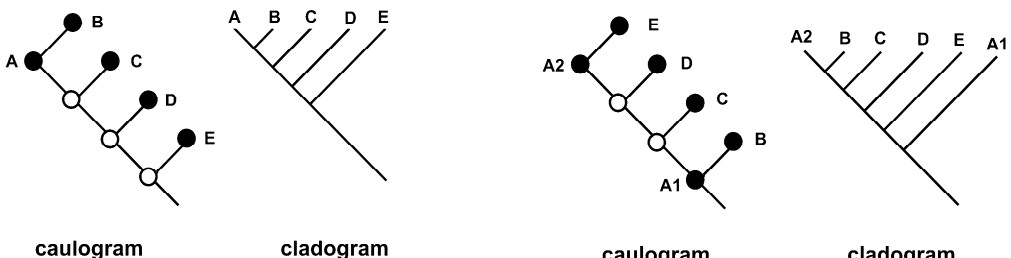

**Figure 3.** Details of microgenera comparing caulograms and cladograms: (**A**) Idealized lineage of four microgenera in-depth, corresponding to beginnings of modern moss flora in the Late Cretaceous. Hollow circles show gradual extinction of species over time, resulting in a tadpole-shaped lineage of structured monophyly. (**B**) Microgenus of optimally one ancestral species (**A**) and four descendant species (**B–D**) as a morphological model for molecular equivalents. (**C–E**) Caulogram and cladogram equivalents showing results of extinction of ancestral molecular strains. (**C**) Surviving ancestral molecular strain at the base. (**D**) Surviving ancestral molecular strain at the apex. (**E**) Two surviving ancestral strains give two molecularly differentiated "cryptic species" A1 and A2.

Positive information deriving from the molecular cladogram may be derived from the mapping of habitat types to various species and the estimation of origination in geologic time of the various subclades. For instance, terminal clades are of arboreal species [46], perhaps adapted to a tree or at least epixylic habitat during the recent Miocene drying of the ecosphere. This matches, to some extent, the largely arboreal recent species of the Streptotrichaceae. With additional studies using microgenera, much further information about the evolution of this large and important group may be gained.

*Chionoloma* Molecular Cladogram

The macrogenus *Chionoloma* Dixon emend. M. Alonso, M. J. Cano & J. A. Jiménez are deconstructed by extracting the microgenera *Oxystegus* (Limpr.) Hilp. and *Pseudosymblepharis* Broth. These remain much the same as classically distinguished. Lumping these genera to fit a classification principle, holophyly, which is not a process in nature, is rejected. *Chionoloma* s.str. is retained, with the addition of New World species [6]. A new combination is needed to include *Chionoloma dubium* (Thér.) M. Alonso, M. J. Cano & J. A. Jiménez in its proper microgenus, and is given below in Section 4.

The cladogram of Alonso et al. [43] is reproduced in part in Figure 2B. A key to *Chionoloma*, as emended by Alonso et al. [44], distinguished the classical genera quite well using morphological traits. The molecular cladogram shows that most species in the classical genera are well distinguished. There are four microgenera: *Oxystegus* (Limpr.) Hilp., *Chionoloma* Dixon s.str., *Pachyneuropsis* H. A. Mill., and *Pseudosymblepharis* Broth. The numbers of species range from one to five, averaging 3.75 species per genus (by chance exactly that of *Syntrichia*). Computations involving small numbers can create disconcerting chance correlations [47].

According to Zander [6], the average number of species in 36 studied microgenera of the moss family Pottiaceae is 3.58, and the mode (the most often recorded) is 4. Apparently, the estimation of the numbers of microgenera in the molecular cladograms of Figure 2 is fairly accurate. Morphological traits also follow a rule of four, that is, generally, a maximum of four descendants. Newly evolved traits, i.e., the novon set, in 36 microgenera were found to average 12.9 per genus, and the mode is 12. There are, thus, about three new traits per species on average. This could not be judged from the molecular cladograms, but further morphological study might confirm this for the estimated microgenera. In a study of a West Indies *Trichostomum* lineage [6], there averaged 3.7 species per genus and 4.0 new traits per species; a partial *Weissia* lineage in the same study averaged 3.8 species per genus and 3.4 new traits per species.

If the optimal number of species in a microgenus is five (ancestor plus descendants), why are there 3.58 to 3.75, as above? In the study of the family Streptotrichaceae [1,2] of 10 microgenera and 30 species, the number of species per microgenus was found to be only 2.8 on average. Half of the genera of the family were monotypic, and if these are deprecated as accumulating remnants of mostly extinct genera, then the average number of species per genus is 4.6. The family Streptotrichaceae is four microgenera in the depth of geologic time (Figure 3A), and extreme age may have generated the unispeciate genera by extinction, while the genera *Chionoloma* s.lat. and *Syntrichia* s.lat. are probably of fewer microgenera in evolutionary depth, thus lacking comparatively large numbers of unispeciate genera. In other words, morphologically studied microgenera averaged 2.8–4.6 species per genus depending on the number of successive, linearly concatenated ancestral species.

There were 4.04 new traits per species overall in Streptotrichaceae, which is acceptable because extinction does not trim novon traits (total traits may be reduced by overall morphological reduction). The estimates of about 3.75 for the average total species per molecular microgenus may ignore ancestral species that may be hidden in the *Syntrichia* molecular cladogram and simply reflect more secondary ancestry (descendants of descendants) in the *Chionoloma* molecular cladogram. This macroevolutionary technique is as yet new, and much further work may explain apparent inconsistencies.

New and important information may be derived from the above molecular study. Because they mutate between generations of descendant species, ancestral species sometimes appear in multiple places in a molecular cladogram. The species that imply ancestral status are labeled in Figure 2B with the letter "A" and a number. *Oxystegus recurvifolius* A1, *O. cylindrotheca* A2, and *O. tenuirostris* A3 contend for ancestral status as all three species appear interspersed. Exemplars of only *O. hibernicus* and *O. minor* appear un-split, and these two species may prove terminal in a caulogram of microgenera. In *Pseudosymblepharis*, *P. angustata* A4 brackets *P. dubia*, *P. orthodontium*, and *P. richardsii* and is thus a good candidate for their shared ancestral species. In a somewhat isolated clade, *O. melanostoma* A5 is a

good bet to be ancestral to *O. arboreus*. In another isolated clade, *P. schlimii* A6 is probably ancestral to *P. perlongifolius*. These signposts are contingent on a thorough morphological analysis, of course.

Another instance of implied granularity in molecular cladograms comes from a study of metadata [5] on several molecular trees that had exemplars of the same species occurring multiple times with exemplars of other species between the nodes. This is molecular paraphyly and was attributed to different molecular strains. The average number of nodes between maximally distant exemplars of the same species was 4.50. This was at first thought to be a lack of resolution, but now it can be suggested that this is evidence of embedded microgenera. The scattered exemplars may be explained as representing one morpho-ancestor with a molecular variant associated with each of two to five descendant species. Extinct molecular strains may be stand-ins for shared molecular ancestors in cladograms (Figure 3B–D), leading to molecularly cryptic "species" as extant ancestral strains A1 and A2 in Figure 3D.

Given millions of years of evolution associated with lineages of microgenera [1,2], ancestor–descendant discrepancies in the geographic range may not negate inferred relationships. Ecosystems and habitats expand and contract over geologic time, resulting in a stew of overlapping sympatry and allopatry.

## 4. New Taxonomic Combination

*Pseudosymblepharis dubia* (Thér.) R. H. Zander, comb. nov. Basionym: *Trichostomum dubium* Thér., Bull. Acad. Int. Géorgr. Bot. 20: 99. 1910.

## 5. Discussion and Conclusions

Molecular systematics has important contributions. One should accept, however, that an ancestral species cannot be identified from molecular data alone. I find that the new genus-level synonymy introduced in the *Syntrichia* study [46] is not justified by new evolutionary information involving ancestor–descendant relationships. In addition, the otherwise well-done classical revision of *Chionoloma* s.lat. by Alonso et al. [44] was poorly served by their prior molecular study [43], which afforded no particular taxonomic guidance. The actual information about possible ancestor–descendant information is well hidden by the imposition of overall holophyly on generically synonymized groups of minimally monophyletic genera.

Minimally monophyletic groups of microgenera may be the reason for the multiplicity of small subclades in molecular cladograms. Microgenera have several interpretive advantages over classical mesogenera or cladistic macrogenera. Examples of lineages constructed from serially arranged and branching minimally monophyletic groups based on morphological data are fully monophyletic. They allow immediate analysis of changes through time by significant expressed traits that may be correlated with environmental changes, and in some cases, adaptational explanations are possible. It has been suggested [48] (p. 101) that because evolution is very complex, there are no easily determined biological laws governing evolution, except possibly Dollo parsimony. The present paper gives evidence that there are complexity-related evolutionary rules that may introduce granularity at several levels of organization. This includes a rule of four descendants being optimal for each ancestral species, leading to a genus concept involving fractal evolution and tadpole-shaped phylogenies reflecting the gradual trimming of taxa by extinction. Lineages composed of microgenera may be self-similar at other scales, with an evolutionary rule of four as optimal for groups of traits per species and genera per family. Potentially adaptive traits may be active sympatrically, and traits that sustain lineages through geological time can be identified. The direct morphological connections of microgenera in caulograms of stem taxa may help provide predictions of floristic change associated with the present environmental crisis.

**Funding:** This research received no external funding.

**Institutional Review Board Statement:** Not applicable.

**Informed Consent Statement:** Not applicable.

**Data Availability Statement:** All data and material used are available in this manuscript.

**Acknowledgments:** The author salutes the Missouri Botanical Garden for its continued support of theoretical work in bryology and evolution.

**Conflicts of Interest:** The author declares no conflicts of interest.

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
