# Peer review of "Minimally Monophyletic Genera Present within Meso- and Macrogenera"

_2673-6500, doi:10.3390/taxonomy4030033_

Round 1

Reviewer 1 Report

Comments and Suggestions for Authors

This work proposes the consideration of small genera as evolutionary units within big genera. Traits defining them can be associated with survival across millions of years of environmental perturbation.

The author analysis the average number of species that composes the clades in well studies genera of the Pottiaceae, such as Syntrichia and Chionoloma, and even in other organisms as Reptilia, and vascular plants, and concluded that they are usually composed of five species, following an evolutionary “rule of four”: four immediate descendant species for each ancestral species.

Although this reviewer is not familiar with the employed method by the author, but with the Pottiaceae famly it seems to me that his conclusion fit very well in the two analysed genera. It would be interesting to see if it could be applied in other macrogenera of mosses, but in affirmative case it would simplified the taxonomy of these genera, as long as they can be morphologicaly distinguished.

Some formal corrections are made in the text.

Author Response

The comments were as edits to the manuscript.

I have made all corrections suggested. The wrong citation is fixed. Confusing sentences are clarified. The new combination is moved.

Reviewer 2 Report

Comments and Suggestions for Authors

Review of:

Minimally Monophyletic Genera Present Within Meso- and Macrogenera

Richard H. Zander 4

The paper addresses to the important issue of the representation in taxonomy the common situations where ancentral and descendant species coexist. The help from the microgenera approach is clearly explained.

For better clearness, I’d suggest to add a short explanation on the conspicuous morphological traits and common putative adaptation from some microgenera, e.g. between Oxystegus tenuirostris-O. cylindricus-group and for Pseudosymblepharis orthodontum-P. angustifolium-group. This may illustrate advances of the microgenera approach, since for most of readers the names in the Fig. 2 would say nothing. Otherwise some of Syntrichia groups may provide some examples.

Text is accurately written, easy and interesting to read.  I found only one typos

line 310             lone   must be long

Author Response

or better clearness, I’d suggest to add a short explanation on the conspicuous morphological traits and common putative adaptation from some microgenera, e.g. between Oxystegus tenuirostris-O. cylindricus-group and for Pseudosymblepharis orthodontum-P. angustifolium-group. This may illustrate advances of the microgenera approach, since for most of readers the names in the Fig. 2 would say nothing. Otherwise some of Syntrichia groups may provide some examples.

Text is accurately written, easy and interesting to read.  I found only one typos

line 310             lone   must be long

RESPONSE: I have fixed the misspelling. The reviewer's suggestion that I detail the trait changes is good, but too complicated for the present paper. I am working on a clarification of the Chionoloma paper that will include traits changes between microgenera. The Syntrichia paper cannot be dealt with without a completely new classical taxonomic study using morphology, and perhaps much more sampling of specimens.